# Glioblastoma and MiRNAs

**DOI:** 10.3390/cancers13071581

**Published:** 2021-03-30

**Authors:** Swalih P. Ahmed, Javier S. Castresana, Mehdi H. Shahi

**Affiliations:** 1Interdisciplinary Brain Research Centre, Faculty of Medicine, Aligarh Muslim University, Aligarh 202002, India; swalihpahmed@gmail.com; 2Department of Biochemistry and Genetics, University of Navarra School of Sciences, 31008 Pamplona, Spain; jscastresana@unav.es

**Keywords:** miRNA, glioblastoma, diagnosis, treatment

## Abstract

**Simple Summary:**

Glioblastoma (GB) is the most common type of malignant brain tumor. It affects 7 per 100,000 people a year. The disease has a poor prognosis and patients generally die within 14 months of diagnosis. Recent studies have reported the crucial role of microRNAs (miRNAs) in clinically-resistant glioblastoma. In this review manuscript we attempted to provide a holistic picture of GB up-regulated and down-regulated miRNAs, in relationship with the expression of other genes, cell signaling pathways, and their role in GB diagnosis and treatment.

**Abstract:**

Glioblastoma (GB) is one of the most common types of lethal brain tumors. Although several treatment options are available including surgery, along with adjuvant chemo and radiotherapy, the disease has a poor prognosis and patients generally die within 14 months of diagnosis. GB is chemo and radio resistant. Thus, there is a critical need for new insights into GB treatment to increase the chance of therapeutic success. This is why microRNA (miRNA) is being potentially considered in the diagnosis and treatment of glioblastoma. The objective of our review is to provide a holistic picture of GB up-regulated and down-regulated miRNA, in relationship with the expression of other genes, cell signaling pathways, and their role in GB diagnosis and treatment. MiRNA treatment is being considered to be used against GB together with radiotherapy and chemotherapy. Moreover, the use of miRNA as a diagnostic tool has also begun. Knowing that miRNAs are isolated in almost all human body fluids and that there are more than 3000 miRNAs in the human genome, plus the fact that each miRNA controls hundreds of different mRNAs, there is still much study needed to explore how miRNAs relate to GB for its proliferation, progression, and inhibition.

## 1. Introduction

Glioblastoma (GB) is the most common and devasting malignant brain tumor [1,2]. It affects 6 per 100,000 people a year [3]. Survival time is on average 2–18 months and a 5-year period of survival occurs only in 5% of patients [4]. Commonly used treatments are surgical resection combined with radiotherapy and chemotherapy [5]. Moreover, radiation exposure and genetic predisposition are well-defined risk factors for glioblastoma. Even, contemporary therapy like radiotherapy combined with the drug temozolomide has provided enhanced survival up to 24 months [6]. The main risk factor of this disease is unclear [7]. MicroRNAs (miRNA) are endogenous small non-coding RNA [8]. Recent studies have reported the crucial role of miRNAs in resistant glioblastoma. Even various miRNAs are involved in the pathogenesis of GB-like contributions in the processes of angiogenesis, metastasis, and tumor growth [9]. MiRNAs regulate gene expression by targeting mRNAs and directly binding to complementary sequences at the 3′UTR of the mRNA [10]. MiRNA then acts as a gene silencer and the miRNA level might regulate the initiation and progression of GB. Many miRNAs have been implicated in GB pathophysiology and therapy [9]. Another interesting study showed that the cytosine methylation of mature miRNA inhibits their tumor suppressive nature and also provides poor prognosis for the glioblastoma [10]. Studies on GB and miRNA are important for improving GB treatment and early diagnosis [9].

## 2. Glioblastoma

Glioblastoma is a WHO grade IV astrocytoma that represents about 30% of all brain tumors [11]. It is a highly vascularized and infiltrating tumor [5]. It originates from glial cells [12] and presents a poor prognosis and high lethality in adults [7]. GB is resistant to the induction of apoptosis, and anti-apoptotic proteins are overexpressed in GB [13]. The fast proliferative ability has been characterized as one typical feature of GB, leading to treatments and dismal prognosis [14]. The current international standard for nomenclature and diagnosis of the WHO (World Health Organization) classified glioma into four classes. More generally into low-grade gliomas (LGG, WHO I and II), and high-grade gliomas (HGG, WHO III and IV). Grade I gliomas relate to lesions that have low proliferative potential and can be cured by surgical procedures, whereas grades II to IV gliomas are highly infiltrative GB and the most common and malignant type of glioma.

## 3. MicroRNAs

MiRNAs are endogenous, small, non-coding RNA molecules containing about 19–24 nucleotides in length [15]. They are also found in plants, animals, viruses, and some bacteria. MiRNAs regulate gene expression by targeting mRNAs, and after directly binding to the complementary 3′UTR site of the target mRNA [3], miRNA acts as an mRNA silencer. MiRNAs can regulate multiple target mRNAs involved in resistance by controlling many biological processes including apoptosis, DNA damage repair, proliferation, cell cycle, senescent, invasiveness, and angiogenesis [16]. Generally, mRNA acts in two possible ways in mRNA silencing: mRNA degradation or translation inhibition through the actions of the RNA-induced silencing complex (RISC). MiRNAs are important regulators of several key signaling pathways implicated in tumor pathogenesis [17]. Some miRNAs can inhibit migration and suppress tumor growth. MiRNA expression levels vary in different tissues and cells and are significantly different in normal cells versus stem cells [18].

## 4. MicroRNA Biogenesis

MicroRNA biogenesis starts with RNA polymerase II binding with DNA in the cell nucleus to form a new primary miRNA. The protein DGCR8 and enzyme DROSHA ribonuclease bind to primary miRNAs. DROSHA cleaves primary miRNA to form a precursor miRNA. The precursor miRNA moves from the nucleus to the cytoplasm with the help of exportin5, to bind Dicer ribonuclease that cleaves precursor miRNA to form a ds-miRNA. RISC (RNA-induced silencing complex) is a multi-protein complex including Argonaute endonuclease that binds double-stranded miRNA and produces single-stranded miRNA. Commanded by RISC, miRNA binds to mRNA. Finally, mRNA degradation or mRNA inhibition of translation is induced (Figure 1).

## 5. MicroRNA in Glioblastoma

An important reason for the inadequate response of GB to radiotherapy is radioresistance. MicroRNAs (miRNAs) are important regulatory molecules that can effectively control glioblastoma by affecting signal transduction pathways [16]. MiRNAs act as diagnostic biomarkers in brain tumors and are used for tumor treatment [12]. MiRNAs mediate the suppression of anti-apoptotic and induce apoptosis in cancer cells [13]. Deregulated levels of cerebrospinal fluid (CSF) in miRNA are associated with glioblastoma. Different types of miRNAs and their expression levels in glioblastoma suggest the feasible development of a new diagnosis method for glioblastoma. Some miRNAs down-regulate the growth of glioblastoma, and therefore, they may be used for therapy against this tumor. On the other hand, as various miRNAs support cell growth in glioblastoma, they might be considered as candidates for the stratification of poor prognosis glioblastoma. Most researchers investigating the relation of glioblastoma and miRNAs have utilized methods to understand the function of miRNAs in glioblastoma. Clinical glioblastoma tissue samples were collected from patients and the samples were frozen at –80 °C. Thereafter, mRNA and protein expression analysis was performed and the methylation profile of the matured miRNAs was also checked. For cell culture, T98G, U251, and U87MG glioma cell lines were grown in Dulbecco’s modified essential medium (DMEM) and supplemented with 10% fetal bovine serum, 1% Penicillin, and 10% streptomycin after culture in a 5% CO_2_ humidified incubator at 37 °C. After reaching 70% confluency, the samples were treated with a miRNA mimic in 6-well culture plates. After 24 h, 48 h, and 72 h cells from each well were trypsinized, single-cell suspensions were prepared, and cells were counted. Cells were centrifuged at 4 °C at 5000 rpm for 5 min. Cell pellets were stored in RNA to be utilized later for further RNA extraction processing. RNA extracted samples were used for cDNA processing. Thereafter, cDNA was used for real-time PCR to determine different gene activity. Moreover, western blotting was also done to determine the protein level, and we then performed statistical analysis of the expression results obtained by real-time RT-PCR. For diagnosis, we collected the cerebrospinal fluid sample, isolated the RNA, used amplification after the qRT-PCR, increasing sensitivity to apoptosis induced by the DNA alkylating agent temozolomide in vitro, and performed western blotting. DS (diagnostic score) values were calculated and used in the diagnosis of the tumor.

### Examples

(1) MicroRNA-342: anti-apoptotic genes BCL2L1 and MCL 1 are silenced by miR-342 activity. BCL2L1 and MCL1 are anti-apoptotic genes that induce GB growth and survival. An increased expression of miR-342 reduced anti-apoptotic gene expression. An overexpression of miR342 led to apoptosis in GB cells. MiR-342 can then be used in therapy against GB, as it may be considered to be a tumor suppressor miRNA [12] (Figure 2).

(2) A microRNA-302-367 cluster efficiently leads to the disruption of glioma-initiating cells and their tumorigenic properties. MicroRNA-302-367 inhibits the CXCR4 pathway that leads to disruption of the Shh-GLI-NANOG network. Then, the miR-302-367 cluster down-regulates GB and can also be considered to act as a tumor suppressor miRNA [5] (Figure 2).

(3) MicroRNA-7-5p inhibits cell migration and invasion in GB through targeting SATB1 (special AT-rich sequence binding protein 1). MicroRNA-7-5p is, therefore, a potential biomarker that might be used for the treatment of glioblastoma [19]. It acts, biologically, as a tumor suppressor miRNA. It appears to be down-regulated in GB samples and cell lines (Figure 2).

(4) Tumor suppressor miRNAs such as miR-34, miR-128, and miR-181 are down-regulated in GB cells. The up-regulation of miR-34 could enhance cellular reactive oxygen species (ROSs), increase radio sensitivity, and inhibit cell viability [16]. MicroRNA-128 overexpression could inhibit GB cell growth and enhance cell radio sensitivity through the down-regulation of BMI1. Finally, miR-181 inhibits Bcl-2 and leads to the down-regulation of glioblastoma [16] (Figure 2).

(5) ExRNA directly promotes the release of cytokines such as tumor necrosis factor-α (TNF-α) or interleukin-6 from immune cells. Analyzed exRNA-loaded microvesicles derived from the CSF of patients with LGG demonstrated that the level of miRNA is higher as compared to controls. Six miRNAs, miR-4443, miR-422a, miR-494-3p, miR-502-5p, miR-520f-3p, and miR-549a are overexpressed in tumors of glial origin. MiR-549a and miR-502-5p expression correlated with the prognosis of the glial tumors. MiRNAs such as miR-4443, miR-422a, miR-494-3p, miR-502-5p, miR-520f-3p, and miR-549a inhibit apoptosis and up-regulate glioblastoma. These miRNAs levels may be used for the diagnosis of glioblastoma [11] (Figure 3)

(6) miR-124, miR-128, and miR-137 are associated with neuronal differentiation and down-regulation of glioblastoma cells. MiR-124-dependent neuronal differentiation reduces glioblastoma aggressiveness. An ERK1/2-miR-124-SOX9 axis regulates neuronal differentiation in GB cells. Neuronal differentiation in response to MEK inhibition is dependent on miR-124 induction in human GB cells. ERK1/2 activation increased miR-124 expression that induced neuronal differentiation in SOX9-expressing GB cells. The overexpression of miR-124 induces neuronal differentiation that abrogates GB aggressiveness [20].

(7) Tetraspanin 17 (TSPAN17) induces abnormal cell growth in GB and is a member of the tetraspanin family of proteins. It may potentially act as an oncogene associated with GB pathology. MiR-378a-3p suppresses the progression of GB by reducing TSPAN17 expression, and may thus serve as a potential therapeutic target for treating patients with GB. A decreased expression of miR-378a-3p and an increased expression of TSPAN17 are associated with glioblastoma. Henceforth, miR-378a-3p down-regulates GB [17] (Figure 2).

(8) MicroRNA-145 is typically a low-expressed miRNA in glioma stem cells (GSCs). The combined effects of miR-145 and dimethoxy curcumin (DMC) are involved in the miR-145/SOX2-Wnt/β-catenin pathway. MicroRNA-145 increased glioma cell apoptosis by inhibiting BNIP3 and Notch signaling and inhibited cell proliferation, adhesion, and invasion by targeting Sox9. The microRNA-145/SOX2-Wnt/β-catenin axis was critical in the DMC-mediated inhibition of GSCs, and the up-regulation of miR-145 could effectively enhance DMC effects on anti-GSCs [18].

(9) MicroRNA-940 showed a low expression in GB cells and glioma tissues. Cyclin kinase subunit 1 (CKS1) is up-regulated in glioma. Correlation analysis indicated that miR-940 expression was inversely related to CKS1. The knockdown of CKS1 significantly induced cell cycle arrest and restrained GB cell proliferation. The overexpression of miR-940 can restrain the proliferative ability of GB cells and inhibit cell cycle progression by directly binding to the CKS1 3′UTR region. MiR-940 functions as a GB down-regulator by directly targeting CKS1 in glioma. MicroRNA-940 inhibits glioma cells proliferation and cell cycle progression by targeting CKS1 [14] (Figure 2).

(10) MicroRNA-21 up-regulates glioblastoma. The extracellular vesicles “exosomes” have emerged as novel and powerful drug delivery systems. The exosomal transfer of miRNAs or anti-miRNAs to tumor cells has provided a new approach. Interestingly, researchers have been able to therapeutically apply miRNAs to combat cancer. The down-regulation of miR-21 expression in glioma cell lines was shown [21,22].

(11) There is a positive correlation between Cdh4 expression levels and glioblastoma cell proliferation. The silencing of Cdh4 is important for inducing a decrease in the infiltrative ability of human glioma cells. Cdh4 silencing was performed by six different microRNAs-expressing retroviral particles (hereinafter named miRcdh4). MicroRNA-Cdh4 transduction induced a strong down-regulation of Cdh4 mRNA in glioma-initiating cells. MicroRNA-Cdh4 down-regulates Cdh4 and also the infiltrative ability of glioblastoma. Cdh4 may be used as a prognostic marker, since its expression is related to a shorter survival time in glioma patients [23] (Figure 2).

(12) MicroRNA-101 is down-regulated in human cancer cell lines such as lung cancer, breast cancer, embryonic rhabdomyosarcoma, laryngeal squamous cell carcinoma, and glioblastoma. Moreover, miR-101 directly targets SOX9, which regulates the proliferation and invasion of glioma cells both in vitro and in vivo [24]. Furthermore, miR-101 directly targets KLF6, thereby inhibiting CHI3L1 expression and blocking the activation of the MEK1/2 and PI3K signaling pathways. Therefore, miR-10 down-regulated glioblastoma could be a potential target for glioblastoma treatment [25] (Figure 2).

(13) MicroRNA-338 is a tumor suppressor which suppresses the PKM2/β-catenin axis, and is down-regulated in GB [26]. Both miR-338-3p and miR-338-5p are differentially expressed in GB and non-tumor brain tissue. Moreover, miR-338-5p with radiation leads to significantly decreased cell proliferation, increased cell cycle arrest, and apoptosis [27]. Therefore, this signified the suppressive role of miR-338-5p in GB. MiR-338-5p even inhibited TSHZ3 expression and the promotion of MMP2 expression. Because miR-338-5p inhibited glioma growth it may therefore be used as one of the diagnostic markers for high-grade gliomas [28]. Notably, miR-338-3p expression is negatively correlated with GB. A low miR-338-3p expression is associated with increased mortality and disease progression in glioblastoma patients. Therefore, miR-338-3p showed clinically relevant tumor suppressing behaviors in GB [29] (Figure 4).

(14) All-trans retinoic acid (ATRA) treatment can modulate miRNA expression patterns. Therefore, it may be used for anticancer therapy. Tumors were treated with ATRA concentrations ranging from 10^−3^ μmol/L to 10^2^ μmol/L for 24 h to 21 days. ATRA treatment was able to modulate more than 300 miRNAs and down-regulated the growth of cancer cells. ATRA seems then to be effective for GB treatment and prevention [30].

(15) MicroRNA-153 has a low-level of expression in glioblastoma. Moreover, miR-153 is a regulator of apoptosis as it reduces the protein levels of the anti-apoptotic Bcl-2 and Mcl-1. Additionally, miR-153 inhibited the expression of Bcl-2 and Mcl-1 by directly targeting the 3′UTR regions of their respective mRNAs. Therefore, miR-153 may be further developed as a promising anti-tumor target candidate [31] (Figure 2).

(16) PRC2 activates many genes indirectly by repressing other repressors. PRC2 directly represses miRNAs, and through this repression of miRNAs PRC2 activates interferon-stimulated genes in GB. PRC2 activates a set of interferon-stimulated genes that are targeted by these miRNAs. This PRC2-miRNA-ISG network is an important regulator of gene expression programs in glioblastoma. PRC2 acts as an oncogene by silencing tumor-suppressive protein-coding genes or non-coding RNAs [32].

(17) The introduction of umbilical cord mesenchymal stem cells (UCMSCs) into glioma cells down-regulates GB growth and development. UCMSCs releases extracellular vesicles (EVs) containing miRNAs which can down-regulate multiple pathways in GB. EVs secreted by UCMSCs contain multiple types of miRNAs and their activity in gliomas will be different.

MicroRNA-199a down-regulates AKT-mTOR pathways leading to the down-regulation of survivin. The further down-regulation of survivin is shown in highly mitotic cells, developing embryos, and cancer. The increased expression of survivin induces aggressiveness in gliomas. MiR-199a inversely correlates with the expression of survivin in GB [33].

The MET gene regulates the signaling system of glioma stem cells by increasing their proliferation and migration rates. MicroRNA-410 transfected into GB results in growth suppression by inhibiting MET expression [34]. OIP5-AS1 expression is higher and miR-410 is lower in glioma tissues. Down-regulation of OIP5-AS1 induces G0/G1 phase cell cycle arrest and the apoptosis of glioma cells. Silencing OIP5-AS1 is blocking the Wnt-7b/β-catenin pathway via targeted up-regulating miR-410, inhibiting growth, invasion, and migration while promoting apoptosis in glioma cells and also down-regulation of glioblastoma [34].

Dysregulated epidermal growth factor receptor (EGFR) signaling in combination with the dual inactivation of INK4A/ARF and PTEN, leads to gliomagenesis. MiR-146a targets the Notch protein, consequently reducing epidermal growth factor receptor (EGFR) expression. MiR-146a presents a low level of expression in glioblastoma [35].

(18) MicroRNA-181 targets cyclin B1 3′UTR and down-regulates its expression in GB. MiR-181b down-regulates KPNA4 in glioblastoma. It leads to a decrease in NF-Kb activation and the down-regulation of GB [33]. MiR-181a and miR-181c bind to Notch2 UTRs, down-regulating its expression. MiR-181c inhibits epithelial-mesenchymal transition (EMT) by down-regulating N-cadherin and vimentin [33]. MiR-181d directly binds to methyl-guanine-methyl-transferase (MGMT) 3′UTR, sensitizing GB to temozolomide. MiR-181d suppresses multiple signaling pathways, including MAPK/ERK and PI3K/AKT pathways, by K-ras activation. Therefore, miR-181d successfully down-regulates glioblastoma [33]. MGMT expression inversely correlates with miR-181d expression in independent glioblastoma cohorts [36].

Serum miR-145-5p levels were significantly decreased in glioblastoma patients. Serum miR-145-5p is a reliable diagnostic marker of glioblastoma. The high miR-145-5p serum group survived significantly longer than the low miR-145-5p serum group [37]. MiR-145 directly targets ADAM19 3′UTR, down-regulating glioblastoma [33].

(19) Temozolomide (TMZ) inhibits glioblastoma cell proliferation by inhibiting the expression of miR-223 and also leads to the increased expression of tumor suppressor Pax6. The overexpression of miR-223 increases TMZ chemoresistance in GB cells [38], and miR-223 is positively correlated with epithelial-mesenchymal transition (EMT). MiR-223 might then be considered as a glioblastoma growth promoter miRNA [39] (Figure 3).

(20) MicroRNA-26a was first identified in colorectal cancer [40]. MiR-26a promotes a decrease in expression of the tumor suppressor genes PTEN and RB1, and decay in MAP3K2/MEKK2 expression. MicroRNA-26a down-regulates ataxia–telangiectasia mutated (ATM) expression by directly targeting the ATM-3′-UTR in GB cells. MiR-26a overexpression also enhances the radio sensitivity of GB cells. This shows miR-26a may be used as a tumor suppressor miRNA against GB [41].

(21) MicroRNA-203 was first identified in bladder cancer. It showed low levels of expression in GB compared to normal brain tissue [42]. Moreover, miR-203 was able to inhibit the proliferation and invasion of GB cells, partially at least via suppressing the expression of PLD2 protein. Therefore, miR-203 down-regulates glioblastoma growth. Hence, it may be used for glioblastoma therapy and diagnosis [43].

(22) MicroRNA-181 is down-regulated in GB. Overexpression of miR-181 inhibited glioblastoma cell proliferation, invasion, and migration, arrested glioblastoma cell cycle in the G1 phase, and induced glioblastoma cell apoptosis. CCL8 was up-regulated in GB cells and was negatively correlated with miR-181 expression. MiR-181 inhibited glioblastoma cell growth and induced apoptosis by directly targeting CCL8. MiR-181may be used as a prognostic biomarker of glioblastoma [44] (Figure 4).

(23) MicroRNA-504 is down-regulated and suppressed tumor proliferation in glioblastoma. MiR-504 overexpression suppressed GB cell migration, invasion, EMT, and stemness activity. MiR-504 is a negative regulator of the Wnt–β-catenin pathway as it directly represses FZD7 expression. MiR-504 could inhibit cell proliferation and promote apoptosis by targeting the FOXP1 (forkhead box P1) in glioma cells [45] (Figure 4).

(24) MicroRNA-152-3p decreased expression has been reported in GB. The overexpression of miR152-3p increased cisplatin sensitivity in GB cells. SOS1 suppressed the cytotoxic effect of cisplatin in GB. SOS 1 is a diguanine nucleotide exchange factor (GEF) for RAS and Rac 1. SOS 1 converts inactive Ras-GDP into active Ras-GTP in EGF (epidermal growth factor)-stimulated cells. SOS 1 promotes cell survival and growth. Transfection of recombinant SOS1 could effectively reverse the increased cisplatin sensitivity induced by miR-152-3p overexpression in GB cells [3]. Thus, miR152-3p acts as a tumor suppressor miRNA in GB (Figure 4).

(25) The expression of miR-212-3p was significantly down-regulated and negatively correlated with serum and glucocorticoid-inducible kinase 3 (SGK3) in GB. MiR-212-3p directly binds to the 3′UTR of SGK3 and inhibits protein expression. MiR-212-3p also suppressed tumor growth in vivo. MiR-212-3p inhibited the proliferation of GB cells by directly targeting SGK3, and could potentially be used as a new therapeutic target for glioblastoma [46] (Figure 4).

(26) MicroRNA-1 expression in GB cells inhibited proliferation and migration. High fibronectin expression in GB correlates with poor patient survival. Moreover, expression of fibronectin was inversely correlated with miR-1 expression. Knockout of fibronectin expression in GB cell lines inhibited proliferation and migration. MiR-1 in GB cell lines directly targets fibronectin, then inhibits the PI3K/Akt pathway. MiR-1 expression inhibits a PI3K/Akt signaling pathway, which was similarly reversed by fibronectin restoration. The miR-1/fibronectin pathway may be a potential target in GB [9] (Figure 4).

Another miRNA, microRNA-365, also inhibits the proliferation and migration of glioblastoma by inhibiting the phosphoinositide-3-kinase regulator subunit 3(PIK3R3) [47] (Figure 4).

(27) MicroRNA-451 was first found in the human pituitary gland in 2005 by Altuvia et al. [48]. It was reported that miR-451 overexpression led to the down-regulation of theAMP-activated protein kinase (AMPK) complex. Moreover, miR-451 is an inhibitor of the AMPK signaling pathway through the direct targeting of CAB39. The AMPK complex plays an important role in the regulation of the balance between the proliferation and invasion of glioma cells [49]. The expression of miR-451 is low in glioblastoma, thereby it acts as a tumor suppressor. Therefore, miRNA-451 may be used as a target for GB treatment [50]. Furthermore, miR-451 can also be used for the diagnosis of GB, nasopharyngeal carcinoma, esophageal cancer, breast cancer, hepatocellular carcinoma, lung cancer, gastrointestinal cancer, pancreatic cancer, renal cell carcinoma, bladder carcinoma, colorectal cancer, osteosarcoma, cervical cancer, and prostate cancer [50] (Figure 4).

(28) MicroRNA-183 positively correlates with glioblastoma. Neurofilament light polypeptide (NEFL) significantly suppressed glioma cell proliferation and is targeted by MiR-183. MiR-183 inhibits NEFL and promotes glioma cell proliferation. MiR-183 is up-regulated in gliomas compared with normal brain tissues.Silencing the expression of miR-183 inhibited cell proliferation, migration, and invasion. Therefore, miR-183 could be a potential target for glioblastoma treatment [51] and a potential biomarker in glioblastoma [52] (Figure 3).

(29) MicroRNA-590-3p was up-regulated in glioma tissues and confers resistance to radiotherapy in glioblastoma cells. The expression of miR-590-3p is higher in high-grade than in low-grade gliomas. Therefore, miR-590-3p is positively correlated with glioblastoma and contributes to the radio-resistance of GB cells by directly targeting immunoglobulin-like domains protein 1 (LRIG1) [53]. It is then considered to be a tumor suppressor miRNA.

(30) Both miR-221 and miR-222 are up-regulated in glioblastoma. They inhibit cell apoptosis in human glioma cells by targeting the pro-apoptotic gene PUMA, which leads to a decrease in BAX expression and increases Bcl2 expression. Therefore, both miRNAs miR-221 and miR-222 could be potential therapeutic targets for GB intervention [54] (Figure 3).

(31) MicroRNA-135b is upregulated in GB and acts as an oncogene by inhibiting tumor suppressor gene GK5 (glycerol kinase 5) in GB [55]. MiR-135b inhibits GSK3β which induces glioma cell proliferation arrest, decreases clonogenicity, and induces apoptotic cell death in both the extrinsic and intrinsic apoptotic pathways. Therefore, GSK3β is an important therapeutic target for gliomas [56] (Figure 3).

(32) WWOX is a tumor-suppressor gene, which inhibits the proliferation of various cancer cells. It was reported that miR-92 down-regulated the expression of WWOX by targeting its mRNA3′UTR. It has been demonstrated that miR-92 upregulates GB. CircMTO1 directly interacts with miR-92 and subsequently serves as a miRNA sponge to upregulate WWOX expression. Henceforth, miR-92 may then be considered as an oncogenic miRNA and be used as a prognostic biomarker of glioblastoma [57] (Figure 3).

(33) MiR-7 is down-regulated in GB and acts as a tumor suppressor gene by repressing the expression of O-linked b-N-acetylglucosamine transferase (OGT), epidermal growth factor receptor (EGFR), and upstream regulators of the Akt pathway, such as IRS-1 and IRS-2. This leads to the inhibition of proliferation, invasion, and migration of GB cells. Therefore, miRNA-7 can be considered as a potential therapeutic target in GB treatment (Figure 5) [58].

(34) The overexpression of miR-362 inhibited the proliferation and metastasis of GB cells. MiR-362 expression was down-regulated in glioblastoma. MiR-362 directly targets mitogen-activated protein kinase 1 (MAPK1). MAPK1 acts as an oncogene and is involved in tumor development. MiR-362 overexpression inhibits MAPK1 which leads to the inhibitory effect of miR-362 on cell proliferation and metastasis in GB (Figure 5) [59].

(35) MiR-15a-5p is up-regulated in GB and acts as an oncogene by targeting tumor suppressor gene cell adhesion molecule 1 (CADM1). MiR-15a-5p inhibits CADM1 which stimulates Akt phosphorylation. Akt, also known as protein kinase B, is an important protein regulating cell proliferation which is over-activated in GB tissues and cells (Figure 5) [60].

(36) The expression of miR-191 is significantly up-regulated in GB tissues and cells. The overexpression of miR-191 promotes human GB cell growth in vivo and in vitro. MiR-191 acts as an oncogene by directly targeting N-deacetyl-ase/N-sulfotransferase 1 (NDST1) and negatively regulating the NDST1. MiR-191 acts as a tumor promoter partly mediated by repressing NDST1 expression in GB development. Therefore, miR-191 could be a potential target for GB treatment and might be used as a potential biomarker in glioblastoma (Figure 5) [61].

(37) The expression of miR-640 is up-regulated in GB tissues and cells. The overexpression of miR-640 promotes growth in human GB cell lines by targeting and suppressing slit guidance ligand 1 (SLIT1). Therefore, the miR-640/SLIT1 axis may be a novel potential therapeutic target for the treatment and diagnosis of GB (Figure 5) [62].

(38) MiR-522-3p is upregulated in GB, promoting the malignant biological behaviors of this tumor by targeting and inhibiting tumor suppressor gene secreted frizzled-related protein 2 (SFRP2) to activate the Wnt/β-catenin signaling pathway. Collectively, miR-522-3p is a novel therapeutic target for GB treatment via regulating the SFRP2/Wnt/β-catenin axis, therefore, providing a feasible direction for the development of new strategies for GB treatment (Figure 5) [63].

(39) MiR-155-3p is up-regulated in GB. MiR-155-3p acts as an oncogene by targeting transcription factor sine oculis homeobox homolog 1 (Six1) and inhibiting Six1-associated pathways. Six1 regulates the development of several organs by inhibiting apoptosis and modulating cell cycle regulators. The inhibition of Six1 leads to the down-regulation of Bax and p21. The down-regulation of Bax and p21 leads to the proliferation, invasion, and migration of GB cells. Therefore, glioma cell resistance to temozolomide treatment is enhanced by miR-155-3p treatment (Figure 5) [64].

## 6. Future Perspectives

In this review manuscript we have attempted to describe various miRNAs which are either directly or indirectly regulating the growth of glioblastoma. MiRNAs have an immense potential in clinical prognosis and diagnosis, as they regulate more than one molecular pathway. The expression or function of miRNAs may serve as potential therapeutic strategies for glioblastoma treatment. The present findings confirm as novel diagnostic tools for a deeper prognostic classification of glioblastoma intrinsic subtypes, based upon the expression profiles of miRNAs, with the belief that such signatures could lead to the improvement of survival rates and provide a potential platform for further studies on the treatment for glioblastoma. Further work is certainly required to disentangle these complexities, and the functional properties of miRNAs need further investigation. Therefore, this study might help improve future molecular therapies targeting glioblastoma. As some miRNAs act as oncogenes and others act as tumor suppressors, miRNAs could hold a great potential for future use as biomarkers for glioma (Table 1).

## 7. Conclusions

These studies indicate that there is a relationship between glioblastoma and miRNAs. Some miRNAs act as anti-apoptotic genes and glioblastoma inhibitors. Some miRNAs are gene silencers of anti-apoptotic genes and inhibit the growth and survival of glioblastoma. Cerebrospinal fluid miRNAs have the potential to be diagnostic biomarkers for brain tumors. The signature of miRNAs can be used for the diagnosis of brain tumors. More than 3000 miRNAs have been discovered in the human genome. The relationship between miRNAs and anti-apoptosis-related genes remains unclear. Henceforth, this study might help improve the molecular diagnosis and treatment of glioblastoma. The function of miRNAs has been primarily studied based on laboratory research, and no large-scale clinical trial research has been undertaken so far. Therefore, a more in-depth study of miRNAs can help improve the effective treatment against glioblastoma.

## Figures and Tables

**Figure 1 cancers-13-01581-f001:**
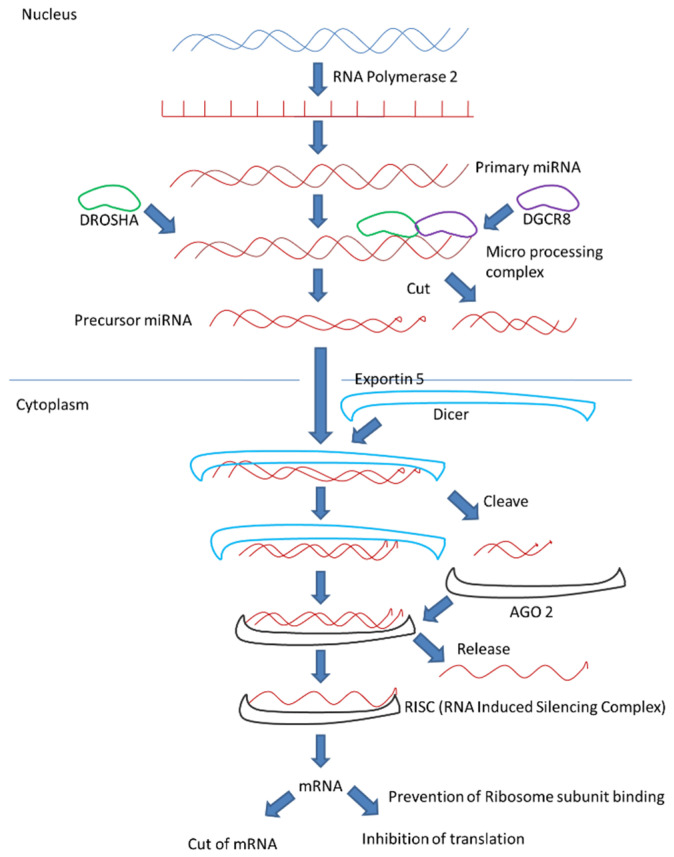
MicroRNA maturation and function: first, synthesis of double strand primary miRNA occurs in the nucleus with the help of RNA polymerase 2. Then primary RNA undergoes modification with the help of a microprocessing complex containing DROSHA and DGCR8 to form precursor miRNA. This precursor miRNA is further transported to cytosol with the help of exportin 5, and thereafter it is cleaved to a smaller fragment. Finally, the smaller double helix miRNA fragment is released to the single helix miRNA and further commanded by the RISC (RNA-induced silencing complex) to degrade or inhibit the target mRNA.

**Figure 2 cancers-13-01581-f002:**
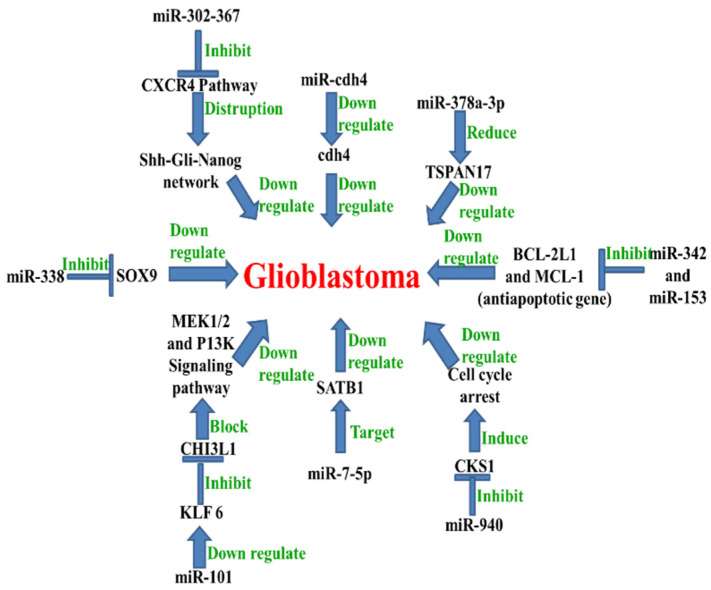
MiRNAs down-regulate glioblastoma: various miRNAs including miR-302-367, miR-Cdh4, miR-378a-3p, miR-342, miR-153, miR-940, miR-7-5P, miR-101, and miR-338 inhibit different cell signaling networks, transcription factors, and anti-apoptotic genes to inhibit glioblastoma growth.

**Figure 3 cancers-13-01581-f003:**
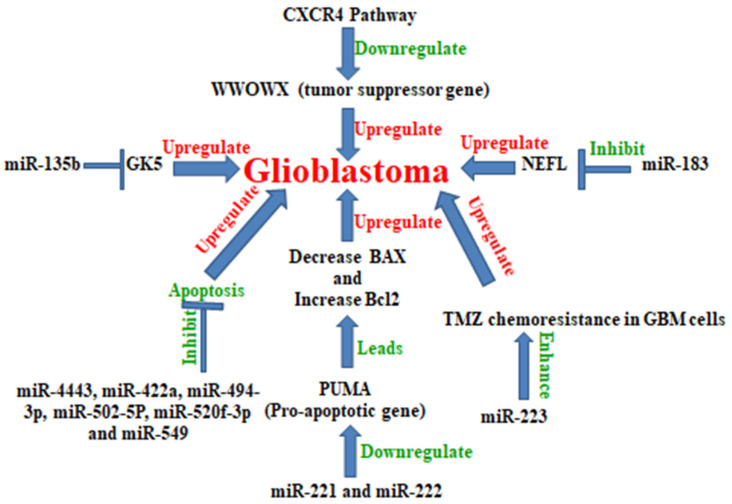
MicroRNAs up-regulate glioblastoma: various miRNAs such as miR-183, miR-135b, miR-221, miR-222, miR-4443, miR-422a, miR-494-3P, miR-502-5P, miR-520f-3p, miR-549, and miR-223 are responsible for the up-regulation of glioblastoma growth.

**Figure 4 cancers-13-01581-f004:**
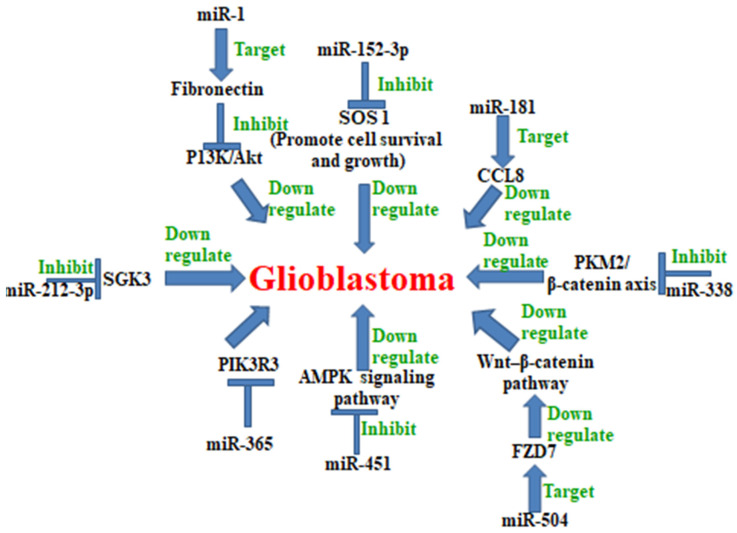
MiRNAs down-regulate glioblastoma: miRNAs such as miR-338, miR-181, miR-152-3p, miR-1, miR-212-3p, miR-451, miR-504, and miR-365 regulate the inhibition of glioblastoma through different pathways including those of the PKM2/β-catenine, P13K/Akt, AMPK (AMP-activated protein kinase), Wnt-β-catenine, and PIK3R3.

**Figure 5 cancers-13-01581-f005:**
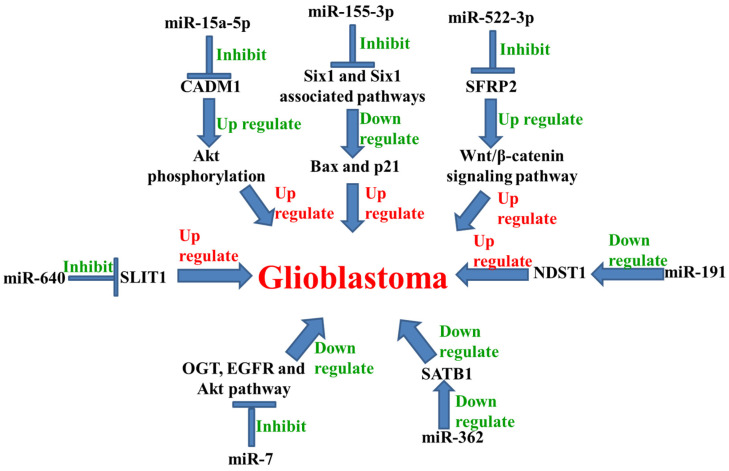
MicroRNA regulation on glioblastoma: various miRNAs such as miR-7 and miR-362 are responsible for the down-regulation of glioblastoma growth. Although, various miRNAs such as miR-15a-5p, miR-155-3p, miR-522-3p, miR-191, and miR-640 are responsible for the up-regulation of glioblastoma growth.

**Table 1 cancers-13-01581-t001:** MiRNAs expression and gene relation in the regulation of glioblastoma.

S. No.	MiRNA Name	Expression in Glioblastoma	Relationship between the miRNA and Its Target Gene	Refs.
1.	miR-153	Down-regulated	↓ Bcl2 and MCL1	[31]
2.	miR-940	Down-regulated	↓ CSK1	[14]
3.	miR-26a	Down-regulated	↓ PTEN and RB1	[41]
4.	miR-203	Down-regulated	↓ PLD2	[43]
5.	miR-338-5p	Down-regulated	↓ TSHZ3 and MMP2	[27,28]
6.	miR-338-3p	Down-regulated	↓ glioblastoma progression	[29]
7.	miR-212	Down-regulated	↓ SKG3	[46]
8.	miR-7-5p	Down-regulated	↓ SATB1	[19]
9.	miR-181a	Down-regulated	↓ CCL8	[44]
10.	miR-128	Down-regulated	↓ BMI1	[16]
11.	miR-124	Down-regulated	↓ MEK	[16]
12.	miR-378a-3p	Down-regulated	↓ TSPAN17	[17]
13.	miR-342	Down-regulated	↓ BCL2L1 and MCL1	[12]
14.	miR-302-367	Down-regulated	↓ CXCR4 and Shh-Gli1-NANGOG	[5]
15.	miR-181	Down-regulated	↓ BCL2	[16]
16.	miR-34	Down-regulated	↓ Enhance ROSs	[16]
17.	miR-137	Down-regulated	↓ Neuronal differentiation	[20]
18.	miR-504	Down-regulated	↓ WNT-β Catenin pathway	[45]
19.	miR-101	Down-regulated	↓ Sox9 And MEK ½	[25]
20.	miR-152-3p	Down-regulated	↓ SOS1	[3]
21.	miR-451	Down-regulated	↓ AMPK signaling	[49]
22.	miR-145	Down-regulated	↓ BNIP3 and Notch signaling	[18]
23.	miR-1	Down-regulated	↓ PI3K/AKT	[9]
24.	miR-199a	Down-regulated	↓ AKT-mTOR	[33]
25.	miR-410	Down-regulated	↓ MET	[33]
26.	miR-146a	Down-regulated	↓ EGFR	[35]
27.	miR-365	Down-regulated	↓ PIK3R3	[47]
28.	miR-7	Down-regulated	↓ OGT, EGFR, and Akt pathway	[58]
29.	miR- 362	Down-regulated	↓ MAPK1	[59]
30.	miR-21	Up-regulated	↓	[21,22]
31.	miR-183	Up-regulated	↓ NEFL	[51]
32.	miR-590-3p	Up-regulated	↑ LRIG1	[53]
33.	miR-221	Up-regulated	↑ PUMA	[54]
34.	miR-222	Up-regulated	↑ PUMA and Bcl2 ↓ BAX	[54]
35.	miR-135b	Up-regulated	↓ GK5	[55]
36.	miR-4443	Up-regulated	↓ Apoptosis	[11]
37.	miR-422a	Up-regulated	↓ Apoptosis	[11]
38.	miR-494-3p	Up-regulated	↓ Apoptosis	[11]
39.	miR-502-5p	Up-regulated	↓ Apoptosis	[11]
40.	miR-520f-3p	Up-regulated	↓ Apoptosis	[11]
41.	miR-549a	Up-regulated	↓ Apoptosis	[11]
42.	miR-92	Up-regulated	↓ WWOX	[57]
43.	miR-223	Up-regulated	↑ EMT	[39]
44.	miR-15a-5p	Up-regulated	↓ CADM1 ↑ Akt	[60]
45.	miR-191	Up-regulated	↓ NDST1	[61]
46.	miR-640	Up-regulated	↓ SLIT1	[62]
47.	miR-522-3p	Up-regulated	↓ SFRP2 ↑ Wnt/β-catenin	[63]
48.	miR-155-3p	Up-regulated	↓ Six1, Bax, and p21	[64]

List of miRNAs responsible for the down-regulation and up-regulation of glioblastoma.

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
