# Peer review of "Glioblastoma and MiRNAs"

_cancers, 2021, doi:10.3390/cancers13071581_

Round 1
Reviewer 1 Report
This review has been improved, while the figures, especially Fig2.3.4 should be better presented.
Author Response
cancers-1139126 reviewers’ comments:
Reviewer 1:
This review has been improved, while the figures, especially Fig 2.3.4 should be better presented.
My major comments are set out below.
Author Response: Thank you. Figures are now inserted next to the text in which they are mentioned and added one new figure named as “Figure 5”. Moreover, we have been also checked minor spelling mistakes in the manuscript.
Reviewer 2 Report
The authors have enriched their paper, which now appears more organized and fluent. However, some aspects need to be further improved before publication in Cancers.
Minor revisions
As requested, the authors have removed the term "multiform" throughout the text, therefore the abbreviation for glioblastoma should be (GB) and not (GBM).
The articles cited are very few (only 51 references), therefore previously I had requested that recent articles, published in 2020, also be included; in the version reviewed by the authors only one article from 2020 was cited, it is too poor, this aspect must be increased by citing other recent articles. I report below some examples:
1) Kong F, Li X, Li S, Sheng D, Li W, Song M. MicroRNA-15a-5p promotes the proliferation and invasion of T98G glioblastoma cells via targeting cell adhesion molecule 1. Oncol Lett. 2021 Feb;21(2):103. doi: 10.3892/ol.2020.12364. Epub 2020 Dec 10. PMID: 33376536; PMCID: PMC7751353.
2) Xue J, Yang M, Hua LH, Wang ZP. MiRNA-191 functions as an oncogene in primary glioblastoma by directly targeting NDST1. Eur Rev Med Pharmacol Sci. 2019 Jul;23(14):6242-6249. doi: 10.26355/eurrev_201907_18443. PMID: 31364126.
3) Alamdari-Palangi V, Amini R, Karami H. MiRNA-7 enhances erlotinib sensitivity of glioblastoma cells by blocking the IRS-1 and IRS-2 expression. J Pharm Pharmacol. 2020 Apr;72(4):531-538. doi: 10.1111/jphp.13226. Epub 2020 Feb 5. PMID: 32026479.
...but there are many others that could and should be mentioned. This would increase the value of the paper.
The paragraph "Future Perspectives" should be inserted after the conclusions, as paragraph 8; this was indicated previously, please correct.
Thank you for adding the paragraphs on abbreviations and the authors contribution.
I had requested to introduce more information in the table (such as relationship between miRNA and its target genes, and references related to each paper) however this was not done, please correct.
Author Response
Reviewer 2:
The authors have enriched their paper, which now appears more organized and fluent. However, some aspects need to be further improved before publication in Cancers.
Author Response: Thank you. We have been further improved the paper.
Minor revisions
As requested, the authors have removed the term "multiform" throughout the text, therefore the abbreviation for glioblastoma should be (GB) and not (GBM).
Authors Response: Thank you for correcting us. We have been changed “GBM” to “GB”
The articles cited are very few (only 51 references), therefore previously I had requested that recent articles, published in 2020, also be included; in the version reviewed by the authors only one article from 2020 was cited, it is too poor, this aspect must be increased by citing other recent articles. I report below some examples:
1) Kong F, Li X, Li S, Sheng D, Li W, Song M. MicroRNA-15a-5p promotes the proliferation and invasion of T98G glioblastoma cells via targeting cell adhesion molecule 1. Oncol Lett. 2021 Feb;21(2):103. doi: 10.3892/ol.2020.12364. Epub 2020 Dec 10. PMID: 33376536; PMCID: PMC7751353.
2) Xue J, Yang M, Hua LH, Wang ZP. MiRNA-191 functions as an oncogene in primary glioblastoma by directly targeting NDST1. Eur Rev Med Pharmacol Sci. 2019 Jul;23(14):6242-6249. doi: 10.26355/eurrev_201907_18443. PMID: 31364126.
3) Alamdari-Palangi V, Amini R, Karami H. MiRNA-7 enhances erlotinib sensitivity of glioblastoma cells by blocking the IRS-1 and IRS-2 expression. J Pharm Pharmacol. 2020 Apr;72(4):531-538. doi: 10.1111/jphp.13226. Epub 2020 Feb 5. PMID: 32026479.
...but there are many others that could and should be mentioned. This would increase the value of the paper.
Author Response: Thank you for this important point. Recent publications have been added. The articles references have been incresed to 64
The paragraph "Future Perspectives" should be inserted after the conclusions, as paragraph 8; this was indicated previously, please correct.
Authors Response: Thank you. We have done as suggested.
Thank you for adding the paragraphs on abbreviations and the authors contribution.
I had requested to introduce more information in the table (such as the relationship between miRNA and its target genes, and references related to each paper) however this was not done, please correct.
Author Response: Thank you. The table has been improved with more information as suggested by the reviewer.
Round 2
Reviewer 1 Report
Figures have been improved now
Reviewer 2 Report
The authors have responded well to all requests, so for me the paper could be accepted for publication in Cancers.
This manuscript is a resubmission of an earlier submission. The following is a list of the peer review reports and author responses from that submission.
Round 1
Reviewer 1 Report
The review article of Swalih et al. investigated the role of miRNAs in glioblastoma. The starting point is interesting, and certainly lays the foundation for further study through preclinical and clinical studies. However, in my opinion, the manuscript needs substantial improvements, both in terms of content and form, before publication in Cancers. Such enhancements should include a more complete literature search, a greater background regarding glioblastoma and miRNAs, and a more accurate exposition of each scientific article mentioned.
My major comments are set out below.
GENERAL
As reported by the WHO guidelines the term glioblastoma multiforme is no longer officially referred to as Glioblastoma; this should be corrected in the manuscript, eliminating the term "multiforme" throughout the text.
Throughout the text the sentences must be more related to each other, thus providing a better connection between the concepts exposed.
ABSTRACT
The abstract exceeds the number of words reported in the guidelines for the authors (the words are 232 instead of 200), it should be redefined, and connect sentences better.
The authors write “A literature search was conducted for GBM and miRNAs at PubMed, with relevant keywords like glioblastoma multiforme, miRNA, diagnosis, treatment, etc., and papers published until 2019 were reviewed” ... this concept is taken for granted, so it could be eliminated from the text;
in addition, why papers published in 2020 have not been included in the review? a good review should contain information as recent as possible, I suggest inserting it in a way that improves the quality of the paper.
1.INTRODUCTION
In the introduction I suggest increasing the background regarding the glioblastoma, and also the miRNAs. Moreover, the sentences must be more related to each other.
Line 35-36: “Glioblastoma is the most common type of lethal brain tumor and the second-most common brain tumor after meningioma” [1]; I did not find this information in the reference entered, check all the references better.
Line 39: Micro RNA is the non-abbreviated form should not be placed in parentheses; must be written MicroRNA (miRNA) and then report only the abbreviation “miRNA” throughout the text.
Line 41: according to what is written in line 39 comment only one abbreviation should be defined in the whole text, therefore only miRNAs should be left and eliminated (ncRNAs).
2.Glioblastoma and 3.WHO grading and Classification could be combined to form a single paragraph
4.MiRNA
Line 62-63: “miRNA are endogenous small noncoding RNA molecules containing about 18-25 nucleotides in length [1]”. Check the reference
6.MiRNA in glioblastoma
Line 99-100: “Some miRNA down-regulate glioblastoma this miRNA is used in glioblastoma treatment. Some miRNA increases and other some miRNA decreases prevent Glioblastoma disease”. The period looks a bit repetitive, rewrite better.
Line 101-112: “Most researchers investigating relation of glioblastoma and miRNA have utilised among the methods used to understand the function of miRNA in glioblastoma are Clinical Patient glioblastoma tissue sample collecting from patients after this are frozen at-80°C. Cell Culture using glioma cell line T98G,U251 and U87 this are cultured in Dulbecco’s modified essential medium (DMEM )and supplemented with 10% Fetal bovine serum, 1% Penicillin and 10% streptomycin after culture in 5% CO2 humidified incubator at 37°C. After added mRNA using vector, after qRTPCR use for amplification. western blotting determine protein level and last statistical analysis of glioblastoma. Diagnosis collect Cerebrospinal fluid sample isolated RNA, after qRT-PCR use amplification, increased sensitivity to apoptosis induced by the DNA alkylating agent temozolomide in vitro, western blotting. DS (Diagnostic score) values are calculated and diagnosis the tumor.” Rewrite everything in a clearer and more comprehensive way
Line 113 and later:
This paragraph represents the main focus of the whole review, the information provided is poor and not very thorough.
For each type of miRNA as many scientific papers as possible (including the most recent) should be reported.
Furthermore, for each work reported as much information as possible should be included; therefore deepen each article analyzed. This would make the paper more useful for the readers and increase its body.
Add an “8. Future perspectives” paragraph.
Add an “Abbreviations” paragraph at the end of the text
Add an “Author’s contribution” paragraph as reported in the guidelines for authors of Cancers
Figures and tables
The figures are complete and well correlated to the paper presented, however Figure 2, Figure 3 and Figure 4 should be inserted next to the text in which they are mentioned.
The table should provide more information. It would be useful to insert more columns including the following points:
- relationship between miRNA and its target genes;
- numbered references, as reported in the text; for example [8] or [9]
References
Verify that references are appropriate with what is reported in the guidelines for authors; in each reference the year of publication should be in bold.
Reviewer 2 Report
This review by Swalih et al. submitted to special issues the 'Recurrent Glioblastoma' aims to provide a holistic picture of GBM up-regulated and down-regulated miRNA, in relationship with the expression of other genes, and their role in GBM diagnosis and treatment. This review has a broad subject and title, however, there are several major issues that the authors need to address.
- The grammar and spelling should be checked thoroughly. There are many misspellings and inconsistent use of English language grammar.
- One major concern is that, as a review send to "Recurrent Glioblastoma" topic, the authors did not review the roles and diagnostic/therapeutic usage in recurrent GBMs. This should be focused on with carefully revision and improvement.
- As I read from line 113 to the end of the review, the authors just list a number of dysregulated miRNA in GBM one by one without getting deeply into details, and many aspects of miRNAs in GBM malignant biology/pathway network/molecular subtype were not discussed. I recommand to re-organize the whole structure of this review.
- fig 2-4 need to be improved.
Reviewer 3 Report
The revised review is a description of the state-of-the-art of miRNAs in glioblastoma. Overall, the authors should broadly and greatly improve the manuscript, for example:
The English language should be edited by a native speaker.
The section 6 “miRNA in glioblastoma” should be rewritten since there are meaningless phrases making it difficult to grasp the sense.
The authors should extend their conclusions, and clarify how clinicians could take profit of the analysis of miRNAs for the different aspects in the clinics, for ex., in preclinical testing, diagnosis (even preventive), prognostics, treatment and clinical follow-up. Which are the potential advantages/disadvantages of considering miRNAs in the different aspects of the clinics? It is feasible the use of these data to improve the current standard-of-care? How real the application of miRNAs is in the clinics? The authors should make a well-thought-through and properly argued position on those and other aspects of the review.
The figures should aesthetically be improved.